# Temporal shifts in dengue epidemic in Guangdong Province before and during the COVID-19 pandemic: a Bayesian model study from 2012 to 2022

Jiaqing Xu[1,2,3☉], Xiaohua Tan[3,4☉], Yi Quan[5,6☉], Dexin Gong[2], Hui Deng[3,4], Jianguo Zhao[2], Xing Huang[2,7], Yingtao Zhang[3,4], Zhoupeng Ren[8], Zuhua Rong[2], Weilin Zeng[2], Xing Li[2], Wenyuan Zheng[2,9], Shu Xiao[2,9], Jianpeng Xiao [ID][1,2,7,9*], Meng Zhang[3,4*]

1 School of Public Health, Guangdong Pharmaceutical University, Guangzhou, Guangdong, China, 2 Guangdong Provincial Institute of Public Health, Guangdong Provincial Center for Disease Control and Prevention, Guangzhou, Guangdong, China, 3 Guangdong Provincial Center for Disease Control and Prevention, Guangzhou, Guangdong, China, 4 Guangdong Workstation for Emerging Infectious Disease Control and Prevention, Guangzhou, Guangdong, China, 5 Pingshan Hospital, Southern Medical University, Shenzhen, Guangdong, China, 6 Pingshan District Peoples' Hospital of Shenzhen, Shenzhen, Guangdong, China, 7 School of Public Health, Southern Medical University, Guangzhou, Guangdong, China, 8 State Key Laboratory of Resources and Environmental Information System (LREIS), Institute of Geographic Sciences and Natural Resources Research, Chinese Academy of Sciences, Beijing, China, 9 School of Medicine, Jinan University, Guangzhou, Guangdong, China

☉ These authors contributed equally to this work
* jpengx@163.com (JX); ccmeng0914@163.com (MZ)

## Abstract

### Background

During the coronavirus disease 2019 (COVID-19) pandemic, the implementation of public health intervention measures have reshaped the transmission patterns of other infectious diseases. We aimed to analyze the epidemiological characteristics of dengue in Guangdong Province, China, and to investigate the temporal shifts in dengue epidemic in Guangdong Province during the COVID-19 pandemic.

### Methods

Based on the data of dengue reported cases, meteorological factors, and mosquito vector density in Guangdong Province from 2012 to 2022, wavelet analysis was applied to investigate the relationship between the dengue incidence in Southeast Asian (SEA) countries and the local dengue incidence in Guangdong Province. We constructed the dengue importation risk index to assess the monthly risk of dengue importation. Based on the counterfactual framework, we constructed the Bayesian structural time series (BSTS) model to capture the epidemic trends of dengue.

### Results

Wavelet analysis showed that the local dengue incidence in Guangdong Province was in phase correlation with the dengue incidence of the prior month in relative SEA countries.

**Data availability statement:** The data used and analyzed during the study was available as Supporting Information (S1 Data).

**Funding:** This work was supported by the Key Area Research and Development Program of Guangdong Province (2022B1111020006 to MZ) and the Natural Science Foundation of China (42071377 to ZPR). The funders had no role in study design, data collection and analysis, decision to publish, or preparation of the manuscript.

**Competing interests:** The authors have declared that no competing interests exist.

The dengue importation risk index showed an increasing trend from 2012 to 2019, then decreased to a low level during the COVID-19 pandemic. From 2020 to 2022, the average annual number of reported imported cases and local cases of dengue in Guangdong Province were 26 and 2, respectively, with a decrease of 95.62% and 99.94% compared to the average during 2017-2019 (594 imported cases and 3,118 local cases). According to BSTS model estimates, 6557 local dengue cases may have been reduced in Guangdong Province from 2020 to 2022, with a relative reduction of 99.91% (95%CI: 98.85-99.99%).

## Conclusion

The incidence of dengue in Guangdong notably declined from 2020 to 2022, which may be related to the co-benefits of COVID-19 intervention measures and the intensified interventions against dengue during that period. Furthermore, our findings further supported that dengue is not currently endemic in Guangdong.

## Author summary

Dengue poses a significant threat to global public health. During the COVID-19 pandemic, the implementation of public health interventions has altered the transmission dynamics of dengue. However, there is a lack of comprehensive studies examining the dengue epidemic throughout the entirety of the COVID-19 pandemic, from 2020 to 2022. Furthermore, limited research has been conducted in countries where dengue is not endemic, such as China. Utilizing surveillance data from Guangdong Province spanning from 2012 to 2022, we developed a dengue importation risk index and captured the epidemic trends of dengue in Guangdong Province. We observed the imported and local dengue cases in Guangdong notably declined during the COVID-19 pandemic, there were only six local dengue cases reported, with no large-scale local dengue outbreaks occurring. The study revealed that the dengue epidemic in SEA may influence the local epidemic in Guangdong, with wavelet analysis showing that imported dengue cases preceded local cases by one month. The BSTS model suggested that 6,557 dengue cases may have been reduced in Guangdong Province from 2020 to 2022, with a relative reduction of 99.91%. We found a significant decline in dengue cases during the COVID-19 pandemic, which may be related to the co-benefits of COVID-19 intervention measures and the intensified interventions against dengue during that period. These further support the notion that dengue is not currently endemic in Guangdong.

## 1 Introduction

Dengue is a systemic viral infection transmitted between humans by Aedes aegypti or Aedes albopictus [1]. The estimates of dengue disease burden in 2013 suggested that 390 million dengue infections occur per year, resulting in 96 million symptomatic infections and 20,000 deaths [2], with transmission occurring in at least 128 countries [3]. Due to modern transportation, urbanization, and global trade, the geographical range of dengue is expanding [4,5]. Dengue epidemics imposed substantial costs on families, health systems, and the economic development of affected countries [6].

On March 11, 2020, the World Health Organization (WHO) declared the global pandemic of the coronavirus disease 2019 (COVID-19) [7]. In response, governments around the world

implemented measures, such as restricting international flights, implementing quarantine and isolation measures, and imposing large-scale lockdowns to reduce the transmission of the virus [8,9]. To prevent imported cases, China implemented strict public health intervention measures in early 2020 [10,11], which reduced the disease burden of other infectious diseases such as influenza, hand-foot-mouth disease and tuberculosis [12]. A study in Sri Lanka found that the risk of dengue transmission was significantly reduced during the lockdown period in 2020 [13]. Cheng et al. found that interventions for COVID-19 may help control the dengue epidemic in 2020 [14]. While previous studies have generally spanned brief periods, predominantly in 2020 or 2021 [15,16], there remains a significant gap in systematic evaluation throughout the entirety of the COVID-19 pandemic from 2020 to 2022. Moreover, the majority of research has concentrated on nations where dengue is endemic [17,18], with a notable absence of studies in countries where the disease is not endemic, such as China.

Guangdong Province has been a region with a high incidence of dengue [19]. From 2005 to 2020, China reported 81,648 local cases of dengue, with Guangdong accounting for 74% of dengue cases in China [20]. In general, the dengue epidemic in Guangdong Province is driven by importation [21,22]. During the COVID-19 pandemic from 2020 to 2022, Guangdong Province strictly implemented the strategy of "dynamic zero-COVID" [23,24]. From 2020 to 2022, the incidence of dengue in Guangdong greatly decreased. Based on surveillance data in Guangdong Province from 2012 to 2022, we aim to further reveal changes in the epidemiological characteristics of dengue and to forecast the epidemic trends of dengue in Guangdong Province from 2020 to 2022. These findings will provide valuable insights for the development of a dengue prevention strategy.

## 2  Materials and methods

### 2.1  Ethics statement

The study was approved by the Ethics Committee of the Guangdong Provincial Center for Disease Control and Prevention (No. W96-027 E-202104).

### 2.2  Data collection

**2.2.1  Dengue data.**  The anonymized data of imported and local dengue cases reported in Guangdong Provincial for 2012–2022 were obtained from the Guangdong Provincial Center for Disease Control and Prevention. Dengue cases were diagnosed according to the National Diagnostic Criteria for Dengue (WS216-2008 and WS216-2018) [25]. Imported dengue cases were defined as those that had been to a dengue-endemic country or region within 14 days before the onset. We obtained weekly dengue case data from six SEA countries that have close contact with China, including the Philippines, Cambodia, Malaysia, Singapore, Viet Nam and Thailand. These data were retrieved from the World Health Organization's Regional Office for the Western Pacific's Institutional Repository for Information Sharing (https://www.who.int/westernpacific/emergencies/surveillance/dengue) and the Thailand Ministry of Public Health (http://www.boe.moph.go.th/boedb/surdata/index.php).

**2.2.2  International flights from SEA into Guangdong.**  The international flights were collected from SEA into Guangdong between 2012 and 2022, using data obtained from the company VariFlight. We used monthly flight data from SEA into Guangdong Province to further delineate the risk of dengue importation.

**2.2.3  Meteorological and Mosquito vector data.**  Meteorological data in Guangdong Province were collected from the China Meteorological Data Science Centre, including temperature, precipitation and relative humidity. Mosquito vector surveillance data, including

the Breteau Index (BI) and Mosquito Oviposition Index (MOI), were obtained from the Guangdong Provincial Center for Disease Control and Prevention.

**2.2.4 public health intervention measures.** The COVID-19 interventions in Guangdong were obtained from official reports, including cases-based measures (quarantine, isolation and health tracking), community measures (maintaining social distancing and regional classification management) and travel-related measures (limited number of international flights). To prevent imported cases of COVID-19, the government restricted the number of international flights on March 12, 2020. Entry management and quarantine were implemented for all immigrants on March 20, 2020. The flight circuit breaker measures were implemented on June 8, 2020. On December 16, 2020, adjustments were made to the weekly suspension of flights. On May 1, 2021, adjustments were implemented regarding the passenger capacity rate of aircraft. Guangdong Province fully lifted restrictions related to the COVID-19 epidemic in January 2023 (Fig 1).

As a high-risk region of dengue epidemic, Guangdong Province paid attention to and even strengthened dengue prevention and control during the 2020-2022 period of COVID-19 pandemic, particularly during the dengue epidemic season. For instance, to prevent the risk of potential dengue importation, Guangdong implemented stricter health surveillance and screening measures for inbound travelers from regions where dengue was endemic, such as enhanced screening for Dengue Virus NS1 antigen. Regular monitoring of mosquito vector density was conducted to identify high-risk areas for vector breeding, facilitating timely vector control and breeding site sanitation. Based on the Joint Prevention and Control Mechanisms against COVID-19 [26], Guangdong strengthened collaboration among multiple departments, including the centers for disease control and prevention, hospitals, police, transportation, and urban management. If new local dengue cases were identified, these mechanisms ensure that

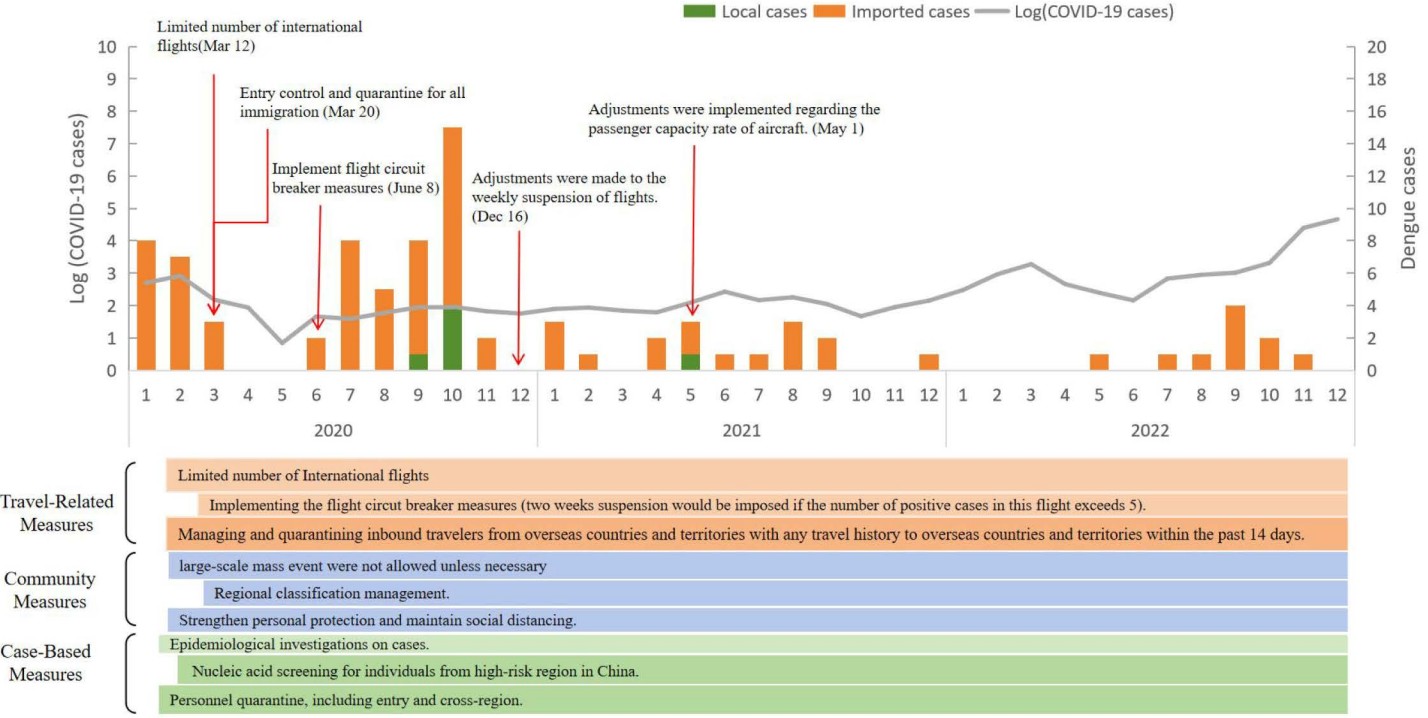

**Fig 1. The epidemic process of COVID-19 and synchronous interventions from 2020 to 2022 in Guangdong Province.**

the prevention and control responsibilities were firmly placed on local government, organizations, departments, and individuals. This led to early case detection and vector control measures, facilitating the immediate control of local dengue outbreaks. Concurrently, health education was intensified to elevate public awareness of dengue prevention, and encourage the public to participate in the patriotic health actions focusing on vector control and mosquito elimination, such as the removal of stagnant water and the reduction of mosquito breeding sites.

### 2.3 Statistic analysis

**2.3.1 Descriptive analysis.** We first described the epidemiological characteristics of dengue in Guangdong Province from 2012 to 2022, and analyzed the number of reported cases over different periods and seasonal prevalence trends. The reported dengue cases were counted in each city in order to identify spatial patterns. Having organized and classified the imported dengue cases, the study then focused on drawing the origin-destination routes of dengue importation from SEA countries into Guangdong.

**2.3.2 Wavelet analysis.** To explore the relationship between dengue incidence in Guangdong Province and SEA countries, wavelet analysis was used to characterize the periodicity of dengue transmission in Guangdong. Phase difference analysis and phase angle diagrams were applied to explore the relationship between the incidence of dengue in Guangdong Province and six selected SEA countries including Cambodia, Malaysia, Philippines, Singapore, Thailand, and Viet Nam.

**2.3.3 Construction of dengue importation risk index.** Based on the number of inbound flights to Guangdong and the dengue incidence in the six SEA countries, the dengue importation risk index was constructed. For the estimation of the counterfactual dengue importation risk from 2020 to 2022, we made the assumption that the volume of inbound flights remained consistent with the levels observed in 2019. This approach allows for an assessment of potential dengue importation risks under a hypothetical scenario where flight traffic was unaffected by the pandemic-related travel restrictions. The dengue importation risk index was estimated according to the study of Lai et al. [27]. The formula is:

$$\mu_{s,m} = I_{s,m} D P_{s,m} \tag{1}$$

$$I_{IMPORT} = \frac{\sum \mu_{s,m}}{n} \tag{2}$$

Where, $\mu_{s,m}$ represents the dengue importation risk index for country s in month m. The $D$ is 10 days [27], which indicates the average latent period and infectious period of dengue, and $P_{s,m}$ represents the monthly number of flights from SEA countries. The $n$ is the number of SEA countries, and $I_{IMPORT}$ represents the dengue importation risk index for six SEA countries.

**2.3.4 Impact of public health interventions on dengue transmission.** This study used the counterfactual framework to construct the Bayesian Structural Time Series (BSTS) model and compared the monthly counterfactual cases with observed cases to forecast the epidemic trends of dengue. The BSTS model consists of three main components: the Spike-and-slab method, Kalman filtering and Bayesian model averaging, specifically defined as follows:

$$Log(y_t) = Z_t^T \alpha_t + \mu_t + \tau_t + \varepsilon_t \quad \varepsilon_t \sim N(0, H_t) \tag{3}$$

$$\alpha_{t+1} = T_t \alpha_t + R_t \eta_t \qquad \eta_t \sim N(0, Q_t) \tag{4}$$

Equation (3) is the observation equation, which links the observed data $y_t$ (the number of dengue cases at month t) with the unobserved latent states $\alpha_t$. In this study, the state vector $\alpha_t$ is composed of trend, seasonal, and regression effects components. $Z_t^T$ is a d-dimensional output vector, $\mu_t$ is the control variables including dengue importation risk index, monthly average temperature, monthly average precipitation, monthly average relative humidity, and mosquito vector density. The above variables were selected based on Spearman cross-correlation analysis to explore local cases and related influencing factors (S1 Fig). In addition, $\tau_t$ is the time trend control variable (long-term trend, seasonal, and lag time of 1-3 months), $\varepsilon_t$ is the observation error. Equation (4) is the transition equation, which defines the change of the latent state $-_t$ over time. $Z_t^T$, $R_t$ and $T_t$ are model matrices that contain unknown parameters and known values (0 and 1). Specifically, $T_t$ is a transition matrix, $R_t$ represents a control matrix and $\eta_t$ means systematic error. We used the Kalman filter to predict the time series, taking into account spikes and slab priors to select the optimal covariates. The Markov Chain Monte Carlo (MCMC) algorithm was employed to simulate the posterior distribution to obtain the final prediction results, and the Bayesian model averaging method was used to smooth a large number of potential models [10,28].

To test the performance of the BSTS model fitness, data from 2012 to 2017 were used for training, and the data from 2018 to 2019 were used for prediction. In this study, we calculated the Mean Absolute Percentage Error (MAPE) and $R^2$ to assess the model fitness. Based on the fitness model, we further predicted the dengue incidence from 2020 to 2022. To investigate the epidemic trends of dengue, we used the relative reduction rate in dengue incidence from 2020 to 2022 as the indicator. Relative reduction (%) = 100% × (number of predicted cases – number of observed cases)/ number of predicted cases.

**2.3.5. Sensitivity analysis.** To assess the stability of the results of the BSTS prediction model, we incorporated two or three meteorological variables (temperature, precipitation and relative humidity) and changing mosquito density variables (BI or MOI) as factors influencing dengue incidence in the BSTS model.

All the analyses and graphical plotting were implemented in R software (version 4.3.1), and R packages "CausalImpact" and "bsts" were used for BSTS model fitness. *P* values (two-tailed) less than 0.05 were considered statistically significant.

## 3  Results

### 3.1  Epidemiological characteristics of dengue in Guangdong

From 2012 to 2022, a total of 62,078 dengue cases were reported in Guangdong Province, including 59,753 (96.25%) local cases and 2,325 (3.75%) imported cases (Fig 2A and 2B). The imported cases were mainly distributed in cities such as Guangzhou, Shenzhen, and Foshan (Fig 2C). Imported dengue cases originated primarily in SEA, particularly in Cambodia, Thailand, Malaysia, Vietnam, Philippines, and Singapore, accounting for 87.56% of all imported cases (Fig 2D). Local cases showed a seasonal trend from 2012 to 2019. The number of imported cases started to increase in 2017, reaching a peak in 2019. However, from 2020 to 2022, the average annual imported cases and local cases in Guangdong Province were 26 and 2, respectively, with a decrease of 95.62% and 99.94% compared to that during 2017-2019 (594 imported cases and 3,118 local cases) (S2 Fig).

### 3.2  The correlation of the dengue incidence between Guangdong Province and relative SEA countries

There were annual variations in dengue incidence in the six SEA countries, showing an upward trend from 2012 to 2019, a downward trend from 2020 to 2021, and a resurgence in

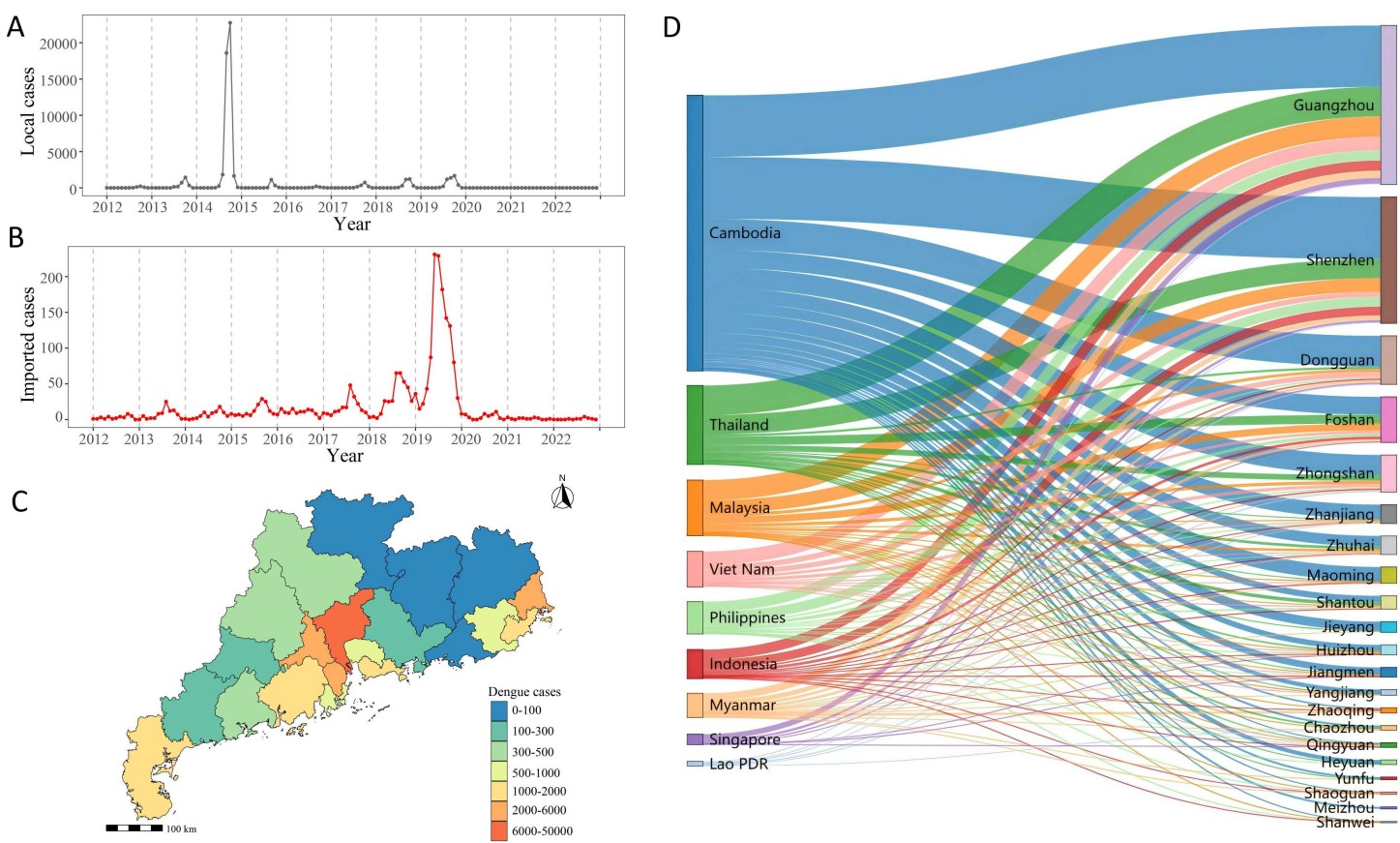

**Fig 2. Epidemiological characteristics of dengue in Guangdong Province from 2012 to 2022.** (A) The monthly number of total local dengue cases. (B) The monthly number of total imported dengue cases. (C) Spatial distribution of local dengue cases. Based map sourced from the Resource and Environmental Science Data Platform (https://www.resdc.cn/DOI/DOI.aspx?DOIID=121) [44]. (D) Origin-destination routes of dengue importation from SEA into Guangdong.

2022 (Fig 3A). From 2012 to 2019, the international flight traffic of SEA countries showed an increasing trend year by year (Fig 3B). The transportation volume experienced significant decreases compared to 2019, dropping by 85.61% in 2020, 97.89% in 2021, and 87.34% in 2022. Wavelet analysis showed that the dengue incidence in Guangdong Province was phase correlated with that in SEA, with a phase difference of about one month (Fig 3C and 3D), which means dengue cases and imported cases in SEA always preceded local cases in Guangdong Province for about one month.

### 3.3 Estimation of the dengue importation risk

As shown in Fig 4A, the dengue importation risk from six SEA countries experienced an overall upward trend from 2012 to 2019, reaching an average monthly value of 0.51. However, the importation risk index increased substantially in 2019, reaching 1.29, which was double the average level of previous years. From the counterfactual predictions, the average importation risk index of dengue was 0.62 from 2020 to 2022 (Fig 4B). While the index decreased significantly to 0.05 (a decrease of 91.93%), after the implementation of public health interventions in Guangdong.

### 3.4 The impact of public health interventions on dengue incidence

The final BSTS model fitness results showed a MAPE of 26.64% and an $R^2$ of 0.89 (S1 Table), suggesting that the BSTS model fit the data well. Fig 5 and Table 1 showed the comparison

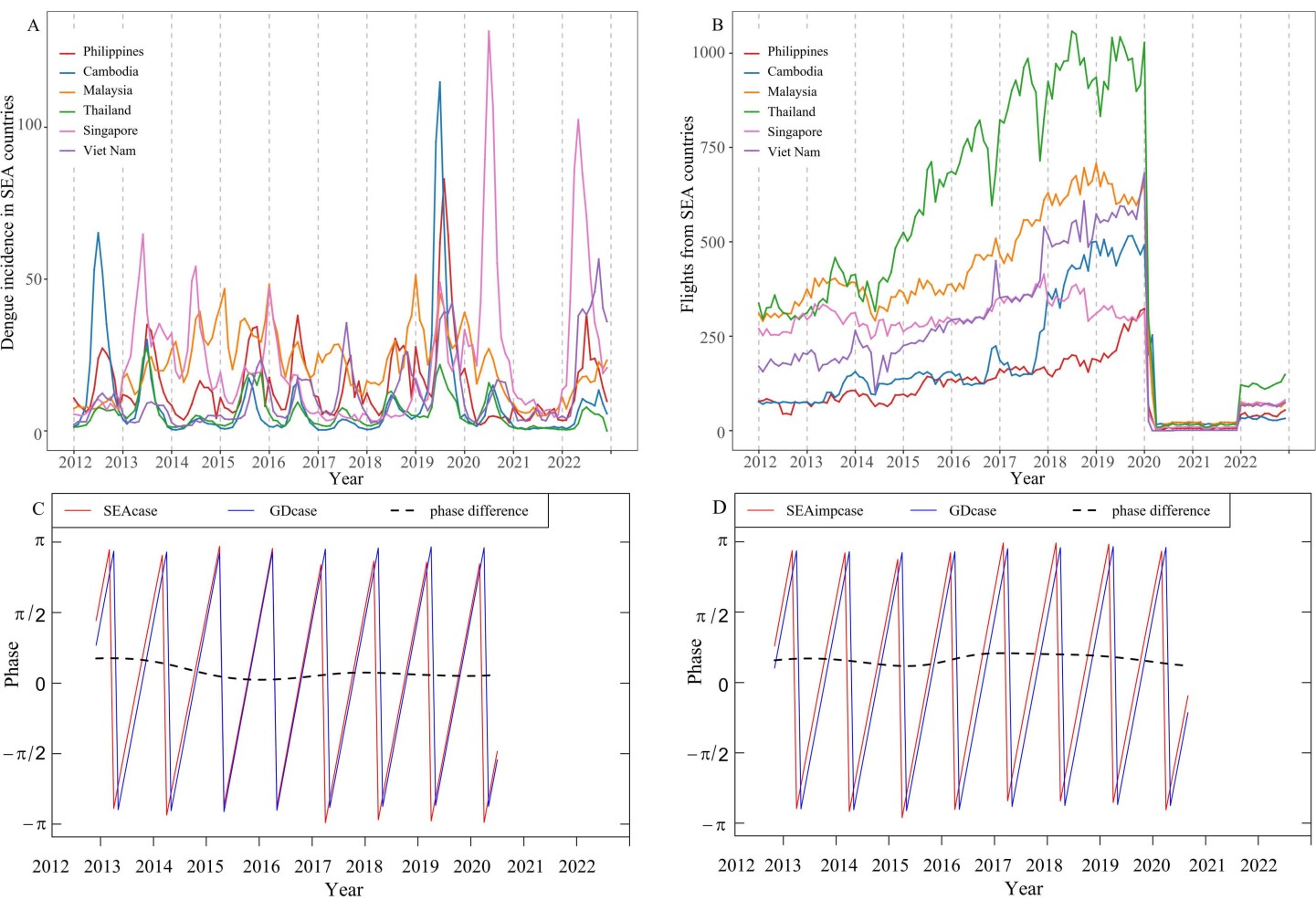

**Fig 3. Temporal distribution and phase relationship of monthly dengue incidence in relative SEA countries and Guangdong Province from 2012 to 2022.** (A) Monthly distribution of dengue incidence in relative SEA countries. (B) Monthly distribution of international flights from SEA countries to Guangdong Province. (C) The phase relationship between local cases and imported cases ("GDcase" represents local dengue cases in Guangdong Province). (D) The phase relationship between local cases and imported cases ("SEAimpcase" represents imported dengue cases from SEA countries to Guangdong Province).

between observed and predicted dengue cases by month during the period from 2020 to 2022. We found that the observed cases were remarkably lower than the predicted counterfactual cases. According to the BSTS model predictions, 6,557 local dengue cases (2289 in 2020, 1894 in 2021, and 2374 in 2022) have been reduced from 2020 to 2022 during the COVID-19 interventions, with a relative decrease of 99.91% (95% CI: 98.85%, 99.99%).

## 3.5 Sensitivity analysis

After adjusting the variables of mosquito vector density (BI or MOI) and relative humidity, the model's stability remained stable. Finally, the model that incorporated three meteorological variables (temperature, precipitation, relative humidity) proved to be more stable than the model that only included temperature and precipitation (S2 Table).

## 4 Discussion

In this study, based on the wavelet analysis and BSTS model estimation in Guangdong Province which has a high risk of dengue transmission, we found that the dengue incidence in

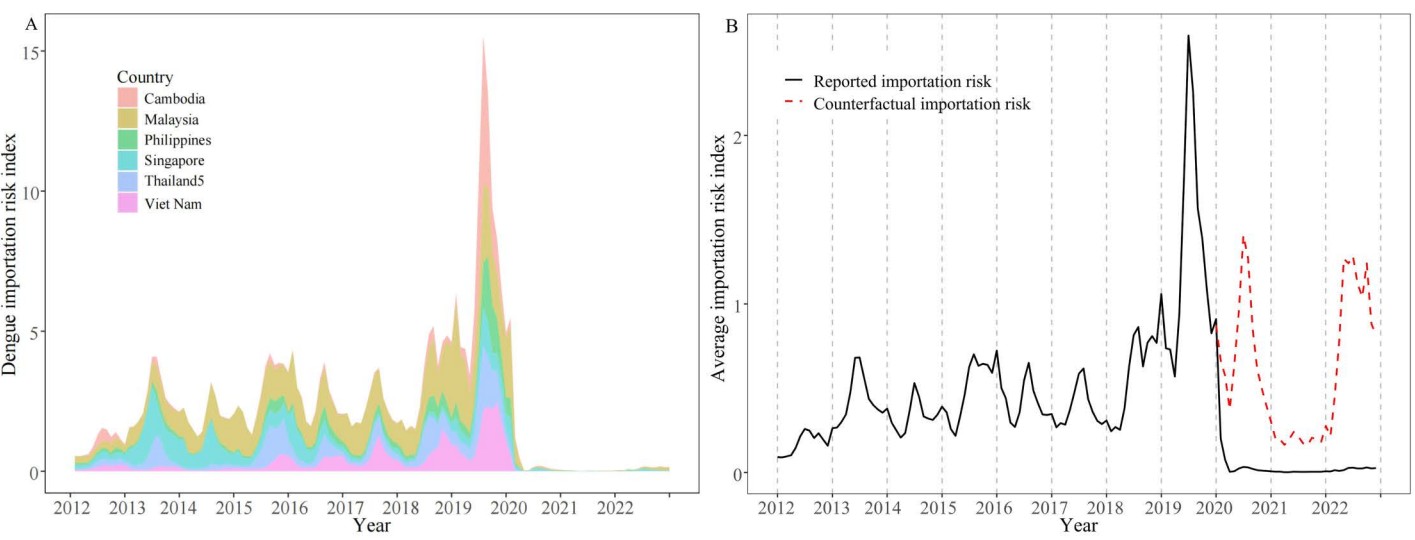

**Fig 4. The dengue monthly importation risks from SEA countries into Guangdong provinces from 2012 to 2022.** (A) The dengue monthly importation risk from six SEA countries. (B) The comparison between monthly predicted counterfactual (red dotted line) and reported importation risk (black solid line) of dengue in Guangdong Province.

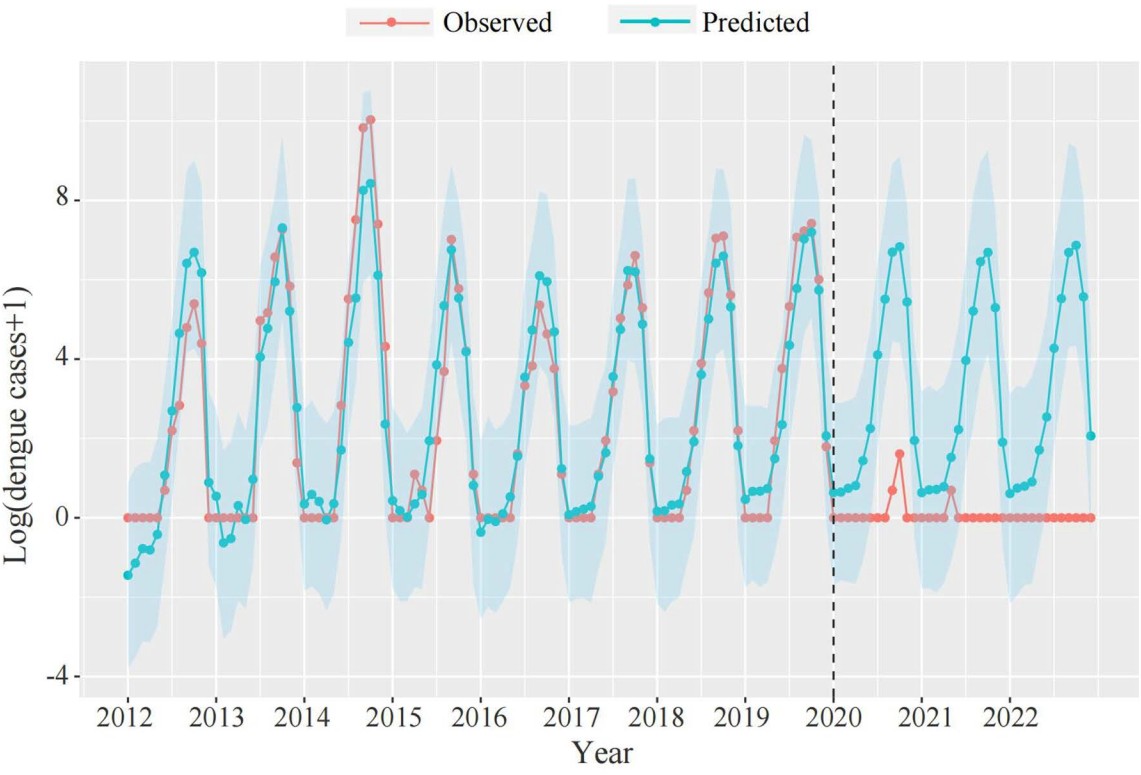

**Fig 5. The comparison between monthly predicted counterfactual dengue cases and observed cases from 2012 to 2022 in Guangdong Province.**

**Table 1.  The relative reduction of dengue incidence in Guangdong Province from 2020 to 2022.**

| Year | Observed local cases | Predicted cases (95% CI) | Averted cases (95% CI) | Relative reduction (95% CI) |
|---|---|---|---|---|
| 2020-2022 | 6 | 6563 (524-78924) | 6557 (518-78918) | 99.91% (98.85%-99.99%) |
| 2020 | 5 | 2294 (214-22290) | 2289 (209-22285) | 99.78% (97.66%-99.98%) |
| 2021 | 1 | 1895 (117-25081) | 1894 (116-25080) | 99.94% (99.14%-99.99%) |
| 2022 | 0 | 2374 (192-31551) | 2374 (191-31551) | 100.00% (100.00%-100.00%) |

Guangdong was closely associated with that in SEA countries. Dengue incidence in Guangdong markedly decreased after the implementation of public health interventions from 2020 to 2022, demonstrating that public health interventions may effectively reduce dengue transmission in Guangdong. The study provides important scientific evidence for dengue prevention and risk assessment.

We observed that the imported dengue cases (87.56%) in Guangdong were mainly from six SEA countries (Cambodia, Thailand, Malaysia, Viet Nam, Philippines and Singapore). The result is consistent with previous studies. Yue et al. showed that 93.9% of imported dengue cases in Guangdong and Yunnan were from SEA countries between 2004 and 2018 [29]. It may be related to globalization and the promotion of the Belt and Road initiative [30,31]. With the increase in economic trade, tourism and cross-border migrant work, the exchanges and cooperation between Guangdong Province and SEA countries have become increasingly close [32]. Wavelet analysis in this study further demonstrated the phase relationship between the dengue epidemic in SEA and the local epidemic in Guangdong Province, and the epidemic occurred one month earlier than that in Guangdong Province. Due to the environmental conditions in Guangdong Province that were conducive to the growth of Aedes mosquitoes, these mosquitoes can infect the virus by biting imported cases and may further bite local residents, potentially leading to the outbreak of local dengue [33,34]. Our study showed that the dengue epidemic in SEA may drive the local epidemic in Guangdong. The study suggested that Guangdong needs to pay close attention to the dengue epidemics in SEA countries during the dengue epidemic season.

In this study, we constructed the dengue importation risk index in Guangdong Province. The dengue importation risk index showed an upward trend from 2012 to 2019, then decreased significantly from 2020 to 2022, dropping from an average of 0.51 (2012–2019) to 0.05 (2020–2022). The result is consistent with previous studies, Tan et al. [35] found that the dengue importation risk intensity in Guangdong Province remained at 3.32 to 9.01 (local cases/ imported cases) from 2016 to 2019, and decreased significantly after 2020 (0.00-0.10). The increase in the risk index from 2012 to 2019 may be associated with modern transportation and global trade [36,37]. During this period, the number of international exchanges in Guangdong Province increased annually, and the dengue importation risk also rose. The importation risk index decreased significantly from 2020 to 2022, which may be attributed to the impact of international flight restrictions during this period [38]. The study suggested strengthening the quarantine and surveillance of travelers returning from dengue-endemic regions.

The BSTS prediction model indicated that from 2020 to 2022, the local observed dengue cases were lower than the counterfactual predicted cases, reaching the lowest level in the past decade. This study found that 6,557 dengue cases may have been reduced in Guangdong Province from 2020 to 2022, with a relative reduction of 99.91%. These results are consistent with the findings of previous studies. Li et al. [39] found that after Yunnan Province implemented entry restrictions, dengue cases decreased by 94.6%. Another study by Surendran et al. [17] found that the movement restrictions led the dengue incidence to decrease by 88% in Sri

Lanka. The significant decline in dengue cases may be related to the immigration management measures implemented in Guangdong Province during COVID-19 pandemic [40]. Specifically, restrictions on international flights and the implementation of flight circuit breakers effectively reduced the risk of importation from dengue-endemic countries such as SEA [39]. Additionally, Guangdong Province implemented social mobility restrictions and case management measures, such as limiting large-scale gatherings, maintaining social distancing, and enhancing health monitoring, which further reduced the probability of local dengue transmission [41]. Our study provide evidence that public health intervention measures such as restricting international flights, strengthening entry management and quarantining for immigration from the dengue epidemic region can reduce the occurrence of dengue. These findings emphasize the important role of public health interventions in controlling the transmission of dengue across regions and provide empirical support for future prevention and control strategies for dengue epidemics.

From 2020 to 2022, the dengue incidence in Guangdong Province decreased significantly. There may be several reasons for this. Firstly, during the COVID-19 pandemic, the implementation of entry restrictions and quarantine measures in Guangdong led to the detection of the vast majority of imported dengue cases at entry ports and during centralized isolation, effectively reducing the risk of dengue importation. Secondly, targeted quarantine and surveillance of inbound travelers from dengue epidemic regions facilitated the timely identification of dengue cases, which likely resulted in a large reduction in transmission. Thirdly, for local dengue cases, the Joint Prevention and Control Mechanisms established during the COVID-19 enabled the immediate implementation of control measures for infectious disease [26], such as dengue case management and vector control. These actions effectively reduced the risk of community transmission of dengue. Additionally, the heightened public health awareness led to greater emphasis on proactive prevention and prompt medical consultation, which was beneficial for the early detection, diagnosis, and treatment of cases, and further reduced the transmission of dengue. Overall, the decline in dengue cases in Guangdong during the COVID-19 pandemic was related to the strengthening of quarantine measures for imported dengue cases and the early detection and intervention of local dengue cases.

Dengue was prevalent in Guangdong Province, and previous studies suggested that it may become endemic in Guangdong [22,42]. We found that Guangdong Province had no reports of continuous local dengue cases throughout the year, showing obvious seasonality, with the peak period from August to November and almost no local dengue cases from January to March [20]. Wavelet analysis revealed that imported dengue cases preceded local cases by one month. The study found that after the implementation of COVID-19 intervention measures in Guangdong Province, imported dengue cases were notably reduced, with only 6 local dengue cases reported (9354 local dengue cases in 2017-2019) and no local outbreaks of dengue occurring during the three years 2020 to 2022. The above research evidence showed that the dengue epidemics in Guangdong Province had been triggered by imported cases. In January 2023, Guangdong Province implemented the "Category B Management" strategy and adjusted public health interventions accordingly [43]. After the resumption of international flights and the removal of entry restrictions, local and imported cases of dengue in Guangdong Province rebounded rapidly, and the number of reported cases in 2023 has returned to the pre-epidemic 2019 level. This phenomenon further supported that the dengue epidemics in Guangdong Province were mainly caused by imported cases.

Our study has several strengths. Based on multi-source data, we quantitatively assessed the impact of public health interventions on dengue transmission. Furthermore, we incorporated influencing factors such as meteorological factors, mosquito vector density, and socioeconomic factors into our model to provide more reliable results. Several limitations should be

acknowledged in our study. First, some dengue cases may have been under-diagnosed and under-reported to some extent from 2020 to 2022, due to the burden of COVID-19 control and prevention. Second, the study included six SEA countries but did not include all SEA countries due to insufficient data, which had a certain impact on the dengue importation risk index. Third, this study is an ecological study design, and the direct effects of specific public health interventions on dengue need further investigation. In addition, dengue transmission may be influenced by complex factors, and although multiple factors were accounted for in the model construction process, there may still be some factors that were not considered.

## 5 Conclusion

During the COVID-19 pandemic, the imported and local dengue cases significantly decreased in Guangdong Province. The study further supported that the dengue epidemics in Guangdong Province were mainly caused by imported cases. The study suggested that timely detection of imported dengue cases and implementation of corresponding public health measures can effectively control the epidemic at the early stage and prevent large-scale outbreaks.

## Supporting information

**S1 Fig.  Correlation analysis between local dengue cases and relative factors.**
(TIF)

**S2 Fig.  Monthly local and imported cases of dengue in Guangdong Province from 2017 to 2022.**
(TIF)

**S1 Table.  BSTS model fitness and predictive effectiveness assessment.**
(DOCX)

**S2 Table.  Sensitivity analysis for BSTS model estimation.**
(DOCX)

**S1 Data.  The data used and analyzed during the study was available as supporting information.**
(XLSX)

## Author contributions

**Conceptualization:** Jianpeng Xiao, Meng Zhang.

**Data curation:** Xiaohua Tan, Yi Quan.

**Formal analysis:** Jiaqing Xu, Dexin Gong, Jianpeng Xiao.

**Funding acquisition:** Zhoupeng Ren, Meng Zhang.

**Investigation:** Xiaohua Tan, Hui Deng, Yingtao Zhang.

**Methodology:** Jiaqing Xu, Dexin Gong, Jianguo Zhao, Jianpeng Xiao, Meng Zhang.

**Project administration:** Jianpeng Xiao, Meng Zhang.

**Resources:** Jianpeng Xiao, Meng Zhang.

**Software:** Jiaqing Xu.

**Supervision:** Jianpeng Xiao.

**Validation:** Jianpeng Xiao, Meng Zhang.

**Visualization:** Jiaqing Xu, Dexin Gong, Xing Huang.

**Writing – original draft:** Jiaqing Xu, Xiaohua Tan, Yi Quan.

**Writing – review & editing:** Jiaqing Xu, Xiaohua Tan, Dexin Gong, Jianguo Zhao, Xing Huang, Zuhua Rong, Weilin Zeng, Xing Li, Wenyuan Zheng, Shu Xiao, Jianpeng Xiao.

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
