## [Decision Letter · Decision Letter 0]

29 Oct 2024

PNTD-D-24-01252Temporal shifts in dengue epidemic in Guangdong Province before and during the COVID-19 pandemic: A Bayesian model study from 2012 to 2022PLOS Neglected Tropical Diseases Dear Dr. Xiao, Thank you for submitting your manuscript to PLOS Neglected Tropical Diseases. After careful consideration, we feel that it has merit but does not fully meet PLOS Neglected Tropical Diseases's publication criteria as it currently stands. Therefore, we invite you to submit a revised version of the manuscript that addresses the points raised during the review process. Please submit your revised manuscript within 60 days Dec 28 2024 11:59PM. If you will need more time than this to complete your revisions, please reply to this message or contact the journal office at plosntds@plos.org. Please include the following items when submitting your revised manuscript:* A rebuttal letter that responds to each point raised by the editor and reviewer(s). You should upload this letter as a separate file labeled 'Response to Reviewers '. This file does not need to include responses to any formatting updates and technical items listed in the 'Journal Requirements' section below.* A marked-up copy of your manuscript that highlights changes made to the original version. You should upload this as a separate file labeled 'Revised Manuscript with Track Changes '.* An unmarked version of your revised paper without tracked changes. You should upload this as a separate file labeled 'Manuscript '. If you would like to make changes to your financial disclosure, competing interests statement, or data availability statement, please make these updates within the submission form at the time of resubmission. Guidelines for resubmitting your figure files are available below the reviewer comments at the end of this letter. We look forward to receiving your revised manuscript. Kind regards, Jin-xin ZhengAcademic EditorPLOS Neglected Tropical Diseases Amy MorrisonSection EditorPLOS Neglected Tropical Diseases

Shaden Kamhawi

co-Editor-in-Chief

Paul Brindley

co-Editor-in-Chief

**Journal Requirements:** **Additional Editor Comments (if provided):** From SE:  Please address both reviewer's comments point by point, but issues raised by reviewer #2 about reframing the objectives of the study should be prioritized.  I also agree that underreporting of dengue cases during the COVID pandemic must be consider.  The role of human movement of dengue virus is very important in dengue virus transmission dynamics, but I would argue that it is social movement between homes that are infested with vectors that may have been affected (family, friends, social contact).   It is very important to describe what control measures (vector control) are normally carried out for dengue control and if they were impacted during the pandemic.   As for Reviewer 1's request for additional time series model, please respond to the request but I would not require that for publication, again focus on the interpretation and implications of your findings and reconsider what are not very practical public health recommendations (see  comments by Review #2).  I agree that one must consider the impact of COVID on vectors. From AE: The study found a significant correlation between dengue incidence in Guangdong and Southeast Asian (SEA) countries, indicating that the dynamics of dengue transmission in the region are interlinked.

But, what specific public health interventions were implemented in Guangdong Province during the COVID-19 pandemic that contributed to the significant decline in dengue cases from 2020 to 2022?**Reviewers' Comments:** Reviewer's Responses to Questions

**Key Review Criteria Required for Acceptance?**

**Methods**

-Are the objectives of the study clearly articulated with a clear testable hypothesis stated?

-Is the study design appropriate to address the stated objectives?

-Is the population clearly described and appropriate for the hypothesis being tested?

-Is the sample size sufficient to ensure adequate power to address the hypothesis being tested?

-Were correct statistical analysis used to support conclusions?

-Are there concerns about ethical or regulatory requirements being met?

Reviewer #1: (No Response)

Reviewer #2: The authors aim to evaluate the impact of public health interventions on the incidence of dengue. However, the statistical methodologies used, such as the reduction in cases by comparing the forecast with the observed cases, do not allow for this level of causal inference. I suggest that the objectives be rewritten to make it clear that the study only evaluates the trend and temporal forecast of cases. In the discussion, I suggest that the authors address the possibility that part of this reduction is likely associated with the health interventions adopted.

**Results**

-Does the analysis presented match the analysis plan?

-Are the results clearly and completely presented?

-Are the figures (Tables, Images) of sufficient quality for clarity?

Reviewer #1: (No Response)

Reviewer #2: As stated above, I believe that the reduction in expected cases based on the observed cases does not allow for the causal inference that the authors aim for in this study, that is, to claim that a 99% reduction in dengue cases is due to the impacts of health interventions adopted since 2020.

Regarding figures and tables, they are well-constructed.

**Conclusions**

-Are the conclusions supported by the data presented?

-Are the limitations of analysis clearly described?

-Do the authors discuss how these data can be helpful to advance our understanding of the topic under study?

-Is public health relevance addressed?

Reviewer #1: (No Response)

Reviewer #2: As stated above, I believe that the reduction in expected cases based on the observed cases does not allow for the causal inference that the authors aim for in this study, that is, to claim that a 99% reduction in dengue cases is due to the impacts of health interventions adopted since 2020.

**Editorial and Data Presentation Modifications?**

Reviewer #1: (No Response)

Reviewer #2: (No Response)

**Summary and General Comments**

Reviewer #1: The authors assessed the temporal shifts in dengue in Guangdong Province before and during the COVID-19 pandemic, using Bayesian structural time series (BSTS) models to forecast transmission trends without public health intervention measures or pandemic. The study found that the transmission patterns of dengue have been markedly reshaped during the COVID-19 pandemic. The findings further supported the notion that the dengue is not currently endemic in Guangdong. Overall, it is well-written and is a useful research contribution.

I have several comments for authors to consider.

1.In the introduction, please add some information about the disease burden of dengue in Guangdong Province.

2. Line 81-83, 390 million infections estimated by WHO should give the time when estimated.

3. In the materials and methods, please add a description of the criteria for identifying "dengue cases"

4. Some teams used GAM or ARIMA model other than BSTS model to predict the incidence of dengue, have the authors tried this model on their data. If so, what is the difference between these two models in prediction？

5. As for this issue, I would like to see the details and the difference of these two models in this study. The author better provide the analysis of GAM or ARIMA model for evaluation and comparison.

Reviewer #2: Discussion

In this study, the authors suggest that the more than 99% reduction in local dengue cases in the studied region was due to public health interventions. They indicate that measures such as restrictions on people’s movement and gatherings, as well as increased health surveillance (lines 344-347), were effective in reducing dengue cases. Additionally, the authors suggest that "restricting international flights, strengthening entry management, and quarantining immigration from dengue epidemic regions can reduce the occurrence of dengue" (lines 347-350). However, over 96% of the dengue cases in the studied region are local, not imported, indicating that drastic measures like air traffic control and immigration quarantines may have some effect, but this effect is likely to be limited, as the vast majority of cases in the studied region were autochthonous. In other words, the authors need to consider this, as the way it was presented suggests that the primary responsibility for the cases in the region lies with the movement of people from outside of China, when in fact it is the opposite.

Moreover, public health interventions are important in controlling dengue; however, it is neither expected nor feasible, from a public health and surveillance standpoint, to achieve a 99% reduction in the incidence of a vector-borne disease, carried by a winged insect highly adapted to the urban environment, merely through measures like controlling movement and gatherings. It cannot be that easy to control dengue, as the study seems to imply. Dengue, like most neglected vector-borne diseases, is much more closely associated with social factors than with a lack of surveillance and health interventions. In other words, it is far more associated with poverty, social inequality, and other chronic structural problems in the societies where it occurs endemically or epidemically. Therefore, it is not plausible to believe that a 99% reduction in incidence occurred because the government controlled people’s movement or something similar. If that were the case, countries like Brazil, which implement important surveillance and health intervention measures such as early diagnosis, immediate treatment, spraying, and health education, would have already achieved some level of success in controlling this disease. However, the opposite is observed, as dengue incidence continues to rise worldwide, even spreading to previously unaffected regions such as Europe. It is also worth noting again that the vector is highly adapted to urban environments and is winged, so while movement restrictions may have some impact, their effect on dengue control is notoriously limited.

Thus, I strongly believe that the more than 99% reduction in local cases since 2020 is due to underreporting, caused by several factors such as low demand for diagnosis and a decreased sensitivity of the local surveillance system in diagnosing and treating potential cases due to the overwhelming burden of COVID-19. This happened globally with various diseases.

In other words, it is important to discuss in more detail that most of the cases in the studied region were local, so suggesting that drastic measures such as air traffic and immigration control, as proposed by the authors as strategies for improving dengue control, need to be reconsidered. Furthermore, the authors need to emphasize that the study contributes by descriptively analyzing the drop in incidence since 2020, but it lacks the causal and epidemiological basis to assert that the causes of this drastic drop were the government’s COVID-related public health intervention measures, such as traffic control and restrictions on gatherings. As I mentioned above, I do not believe it would be that easy to control dengue.

PLOS authors have the option to publish the peer review history of their article (what does this mean? ). If published, this will include your full peer review and any attached files.

**Do you want your identity to be public for this peer review?** For information about this choice, including consent withdrawal, please see our Privacy Policy .

Reviewer #1: No

Reviewer #2: **Yes: ** FÁBIO RAPHAEL PASCOTI BRUHN

---

## [Decision Letter · Decision Letter 1]

9 Jan 2025

Dear Dr. Xiao,

We are pleased to inform you that your manuscript 'Temporal shifts in dengue epidemic in Guangdong Province before and during the COVID-19 pandemic: A Bayesian model study from 2012 to 2022' has been provisionally accepted for publication in PLOS Neglected Tropical Diseases.

Best regards,

Jin-xin Zheng

Academic Editor

Amy Morrison

Section Editor

Shaden Kamhawi

co-Editor-in-Chief

Paul Brindley

co-Editor-in-Chief

Reviewer's Responses to Questions

**Key Review Criteria Required for Acceptance?**

**Methods**

-Are the objectives of the study clearly articulated with a clear testable hypothesis stated?

-Is the study design appropriate to address the stated objectives?

-Is the population clearly described and appropriate for the hypothesis being tested?

-Is the sample size sufficient to ensure adequate power to address the hypothesis being tested?

-Were correct statistical analysis used to support conclusions?

-Are there concerns about ethical or regulatory requirements being met?

Reviewer #1: (No Response)

Reviewer #2: Dear all, in this new version the authors have met the previous requests. Furthermore, they have adequately answered the questions raised.

**Results**

-Does the analysis presented match the analysis plan?

-Are the results clearly and completely presented?

-Are the figures (Tables, Images) of sufficient quality for clarity?

Reviewer #1: (No Response)

Reviewer #2: results aligned with the methodology.

**Conclusions**

-Are the conclusions supported by the data presented?

-Are the limitations of analysis clearly described?

-Do the authors discuss how these data can be helpful to advance our understanding of the topic under study?

-Is public health relevance addressed?

Reviewer #1: (No Response)

Reviewer #2: The conclusions meet the objectives in this new version.

**Editorial and Data Presentation Modifications?**

Reviewer #1: (No Response)

Reviewer #2: Accept

**Summary and General Comments**

Reviewer #1: (No Response)

Reviewer #2: Dear all, in this new version the authors have met the previous requests. Furthermore, they have adequately answered the questions raised.

PLOS authors have the option to publish the peer review history of their article (what does this mean? ). If published, this will include your full peer review and any attached files.

**Do you want your identity to be public for this peer review?** For information about this choice, including consent withdrawal, please see our Privacy Policy .

Reviewer #1: No

Reviewer #2: **Yes: ** Fábio Raphael Pascoti Bruhn

---

## [Editor Report · Acceptance letter]

Dear Dr. Xiao,

We are delighted to inform you that your manuscript, "Temporal shifts in dengue epidemic in Guangdong Province before and during the COVID-19 pandemic: A Bayesian model study from 2012 to 2022," has been formally accepted for publication in PLOS Neglected Tropical Diseases.

Best regards,

Shaden Kamhawi

co-Editor-in-Chief

Paul Brindley

co-Editor-in-Chief
